# A VersaTile Approach to Reprogram the Specificity of the R2-Type Tailocin Towards Different Serotypes of *Escherichia coli* and *Klebsiella pneumoniae*

**DOI:** 10.3390/antibiotics14010104

**Published:** 2025-01-18

**Authors:** Dorien Dams, Célia Pas, Agnieszka Latka, Zuzanna Drulis-Kawa, Lars Fieseler, Yves Briers

**Affiliations:** 1Department of Biotechnology, Ghent University, Valentin Vaerwyckweg 1, 9000 Gent, Belgium; dorien.dams@ugent.be (D.D.); celia.pas@ugent.be (C.P.); agnieszka.latka@ugent.be (A.L.); 2Department of Pathogen Biology and Immunology, University of Wroclaw, Przybyszewskiego 63, 51-148 Wroclaw, Poland; zuzanna.drulis-kawa@uwr.edu.pl; 3Institute of Food and Beverage Innovation, Food Microbiology Research Group, Zurich University of Applied Sciences (ZHAW), Einsiedlerstrasse 35, 8820 Wädenswil, Switzerland; fiee@zhaw.ch

**Keywords:** tailocin, phage tail-like bacteriocin, R2 pyocin, *Klebsiella pneumoniae*, *Escherichia coli*, receptor-binding protein, tailspike, VersaTile, phage–host interactions

## Abstract

**Background:** Phage tail-like bacteriocins, or tailocins, provide a competitive advantage to producer cells by killing closely related bacteria. Morphologically similar to headless phages, their narrow target specificity is determined by receptor-binding proteins (RBPs). While RBP engineering has been used to alter the target range of a selected R2 tailocin from *Pseudomonas aeruginosa*, the process is labor-intensive, limiting broader application. **Methods:** We introduce a VersaTile-driven R2 tailocin engineering and screening platform to scale up RBP grafting. **Results:** This platform achieved three key milestones: (I) engineering R2 tailocins specific to *Escherichia coli* serogroups O26, O103, O104, O111, O145, O146, and O157; (II) reprogramming R2 tailocins to target, for the first time, the capsule and a new species, specifically the capsular serotype K1 of *E. coli* and K11 and K63 of *Klebsiella pneumoniae*; (III) creating the first bivalent tailocin with a branched RBP and cross-species activity, effective against both *E. coli* K1 and *K. pneumoniae* K11. Over 90% of engineered tailocins were effective, with clear pathways for further optimization identified. **Conclusions:** This work lays the groundwork for a scalable platform for the development of engineered tailocins, marking an important step towards making R2 tailocins a practical therapeutic tool for targeted bacterial infections.

## 1. Introduction

Bacterial infections have re-emerged as a significant threat to public health in the 21st century [1]. Given the global spread of pathogens with increasingly accumulated antibiotic resistance mechanisms, the scientific community is compelled to develop alternative treatment therapies [2]. A potential class of novel antibacterials is phage tail-like bacteriocins (PTLBs), also referred to as tailocins [3]. Tailocins are high-molecular-weight (>10^6^ Da), non-replicative protein complexes that share structural similarities with tailed bacteriophages but lack a capsid and a genome. This distinguishes them from phages, which are replicative entities [4]. Upon induction of the SOS response, tailocins are intracellularly synthesized by bacteria and subsequently released to kill competing bacteria within the same niche, providing resistant sister cells with a selective advantage for colonization. The presence of tailocins, initially discovered in *Pseudomonas aeruginosa*, has been reported in many other bacterial genera including *Yersinia*, *Pragia*, *Budivicia*, *Kosakonia*, *Clostridium*, *Xanthomonas*, *Burkholderia*, *Pectobacterium*, *Brevibacillus Listeria*, and *Allorhizobium* ([5,6,7,8], reviewed in [9]). In addition, tailocins are likely to have a much wider distribution and diversity among the bacterial taxa, as bioinformatic studies have identified thousands of other phage-tail-related particle sequences [10,11,12,13]. Morphologically and functionally, tailocins can be classified into R- and F-type tailocins [4].

R-type tailocins resemble the contractile injection system of a T-even phage with a myovirus morphology and consist of an inner tubular core surrounded by an external sheath, as well as a baseplate to which receptor-binding proteins (RBPs) are anchored [14]. R-type tailocins kill their target cell by binding to the cell surface, inducing structural changes in the baseplate of the tailocin. Subsequently, these changes trigger external sheath contraction, driving the inner tubular core to penetrate the bacterial cell envelope. As R2-type tailocins lack a capsid, they form a direct channel between the interior of the cell and the external environment, leading to uncontrolled ion leakage and the collapse of the membrane potential. Consequently, membrane dissipation compromises essential cellular processes, ultimately leading to cell death. F-type tailocins, in contrast, resemble the non-contractile tail of a lambda phage, having a siphovirus morphology with a different, but unknown killing mechanism. Some bacteria, especially *Pseudomonas aeruginosa*, produce both R- and F-type tailocins under the control of the same regulatory genes and are released via the same lysis genes [4,14].

Similarly to phages, the host spectrum of R-type tailocins is determined by its RBP, which recognizes and binds a receptor located on the bacterial surface (e.g., polysaccharides or outer membrane proteins). RBPs are typically homotrimers with a modular structure comprising a conserved N-terminal domain responsible for anchoring to the phage tail or tailocin baseplate, and a variable C-terminal receptor-binding domain (RBD) responsible for host surface receptor recognition [15]. The high amino acid sequence variability observed in the RBD of RBPs mostly enables precise subspecies-specific recognition, leading to a distinct killing spectrum for each tailocin. The subspecies target specificity of tailocins also implies that the target pathogen should be well known prior to successful tailocin application, or that tailocin cocktails should be created to cover a variation in targeted strains. These two strategies are similar to the phage therapy field, namely a personalized (*sur mesure*) approach implying a customized phage selection, or a stable and widely distributed phage cocktail (*prêt-à-porter*) approach [16]. The use of narrow-spectrum antibacterials is increasingly considered beneficial due to their microbiota-friendly nature and the eminent role of a balanced microbiome for human health [17]. There is thus a need to create a large collection of custom tailocins for therapeutic purposes.

Recent advancements in RBP engineering have highlighted the possibility of adjusting the host range of phages and tailocins by substituting complete RBPs or RBDs with their equivalents, originating from different phages, prophages, or tailocins, into conserved viral or tailocin scaffolds [18,19,20,21,22]. The R2 tailocin produced by *Pseudomonas aeruginosa*, also named R2 pyocin, is the best-studied R-type tailocin scaffold and has demonstrated its capacity to accept heterologous C-terminal RBDs with both tail fiber and tailspike characteristics originating from different phages [4]. As such, the antimicrobial range of the R2 tailocin has been successfully reprogrammed across species borders towards *Yersinia pestis* [23], *Escherichia coli* [23,24,25,26], *Salmonella enterica* [27], and *Campylobacter jejuni* [28]. To become a broadly applicable solution of customized, narrow-spectrum antibacterials, either for *sur mesure* or *prêt-à-porter* applications, the R2 tailocin engineering approach should be turned into a convenient platform for the rapid creation of tailor-made tailocins. Until now, studies are limited to only one or a few RBPs, underscoring that tailocin RBP engineering has not yet matured into an easily scalable approach. To produce modified R2 tailocins, the *P. aeruginosa* host strain PAO1 Δ*prf15*, encoding the full operon for the biosynthesis of the R2 tailocin but deficient in its RBP gene, is complemented in trans with an expression vector producing a chimeric RBP. This chimeric RBP is composed of the native R2 N-terminal anchor to attach the chimeric RBP to the tailocin particle and a C-terminal RBD recognizing the receptor of interest. In previous work, tailocin RBP engineering relied on restriction–ligation reactions with traditional type IIp restriction enzymes (REs) to create RBP fusions in expression vectors, which is a cumbersome and tedious approach.

This study aims to introduce a more flexible and convenient screening approach as an initial step to scale up RBP grafting in generalized scaffolds by implementing the VersaTile technique [29]. VersaTile is a combinatorial DNA assembly technique that is convenient for the assembly of non-homologous sequences of modular proteins such as RBPs. It implies two steps: (I) creating a repository of so-called ‘tiles’, consisting of a DNA sequence of interest flanked with location-specific position tags; (II) assembling all tiles in the appropriate vector in a one-step restriction–ligation reaction using only one type of IIs RE [29]. Moreover, all tiles can be easily reused to assemble different constructs in various vectors and thus many scaffolds, allowing a better scalable plug-and-play, Lego-like approach.

As proof of the concept of a tailocin platform with pluggable RBPs, we have grafted RBPs recognizing the O-antigen of *E. coli* and the K-antigen (or capsule) of *E. coli* and *Klebsiella pneumoniae* in the R2 tailocin scaffold. These ESKAPEE pathogens were chosen because of their escalating multidrug resistance and their virulence [30]. The variety of exopolysaccharide structures, namely O- and K-antigens of *E. coli* and *K. pneumoniae*, generally correlates well with the diversity of RBPs produced by the phages infecting *E. coli* and *K. pneumoniae*, respectively. Furthermore, numerous RBPs contain a specific polysaccharide-depolymerizing domain cleaving O-antigen or degrading capsule [31,32,33]. While the O-antigen of *E. coli* has been targeted previously by engineered tailocins, the capsule and the *Klebsiella* species have not been targeted before. Additionally, we demonstrate that a functional bivalent tailocin with a more complex, branched RBP structure targeting two receptors from different species can be created, whereas the natural R2 tailocin of *P. aeruginosa* contains only a single RBP.

## 2. Results

### 2.1. A Plug-and-Play R2 Tailocin Engineering Platform

The VersaTile DNA assembly technique was implemented into the RBP engineering of the tailocin engineering platform, allowing the rapid and straightforward engineering of chimeric RBPs. The engineered RBP constructs consist of two types of tiles. First, the anchor tile originates from the native R2 tailocin and is flanked by two codon-long position tags, namely P_start_ and P_mid_. Second, the RBD tile originates from a wide range of *Escherichia* and *Klebsiella* phage RBPs targeting O-antigen or K-antigen (capsule) and is flanked by position tags P_mid_ and P_end_. These position tags enable the easy and standardized recombination of the two types of tiles in the correct order into the VersaTile-adjusted *P. aeruginosa*-*E. coli* shuttle expression vector pVTD29, used for in trans expression of the chimeric RBP (Figure 1).

Three positive controls were implemented. (I) The native R2 tailocin (R2-WT) in which the RBP was expressed from the *P. aeruginosa* PAO1 genome. (II) The native R2 tailocin with in trans expression of the RBP (R2-WT-trans). Here, the tailocin scaffold is produced by the *P. aeruginosa* strain PAO1 Δ*prf15*, a host with a deficiency in gene *prf15* encoding the RBP. The wild-type RBP is provided in trans using the expression vector pVTD27 without a *lac* repressor, or pVTD29 with a *lac* repressor. (III) The native R2 tailocin with in trans expression of the VersaTile-assembled RBP (R2-WT-VT). The RBP coding sequence has been reconstructed with the VersaTile DNA assembly technique, which introduces (similar to traditional Type IIp restriction enzymes) a six-nucleotide junction between the anchor and RBD. Additionally, a negative control was co-expressed, namely the R2 tailocin lacking the RBP, produced by *P. aeruginosa* Δ*prf15* (R2Δ*prf15*).

### 2.2. Validation of the R2 Tailocin Production Platform Through the Production of Four R2 Tailocin Controls

All R2 tailocin wild-type derivatives mentioned above were enriched using the ammonium sulfate (AS) precipitation method. A two-fold dilution series starting from 300 µg/mL was tested. First, the native R2 tailocin (R2-WT) exhibited clearance at an estimated concentration as low as 0.02 µg/mL (Figure 1E,F). Transparency of the spots decreases with lower R2 tailocin concentrations. Next, R2-WT-trans and R2-WT-VT were evaluated to assess the in trans expression efficiency of the RBP using the VersaTile-adapted vector and the effect of the six-nucleotide junction between the anchor tile and the RBD tile (P_mid_). Here, opaque spots were observed up to a concentration of 1.2 µg/mL for R2-WT-trans and 9.4 µg/mL for R2-WT-VT. This suggests a 60-fold lower efficiency when the R2 tailocin is expressed in trans along with an additional 8-fold reduction in tailocin activity due to the presence of the junction. It is important to note that R2Δ*prf15* also resulted in an opaque spot in the spotting assay at a high concentration of 300 µg/µL, which may be attributed to other compounds in the processed cleared lysate, such as any other protein, mitomycin C, or F-type tailocins produced under the same regulatory genes of the SOS response triggered by mitomycin C [34].

### 2.3. Confirmation of R2-WT, R2-WT-Trans, and R2-WT-VT Activities by Quantitative Survival Assays

R2-WT, R2-WT-trans, and R2-WT-VT showed dose-dependent killing (Figure 2, starting from 0.8, 1.6, and 50 µg/mL, respectively. An estimated dose of 100 µg/mL of R2-WT, R2-WT-trans, and R2-WT-VT reduced the cell count with 6.6 ± 2.0, 4.5 ± 0.6, and 2.5 ± 1.4 log, respectively. These data demonstrate that functional tailocins are successfully produced using the VersaTile-compatible tailocin engineering platform, emphasizing its potential for rapid, convenient, and high-throughput screening. Yet, reductions in efficacy must be taken into account for in trans expression, likely due to an imbalanced production (in amount or time) of the RBP versus the tailocin scaffold, impacting tailocin assembly and the proportion of correctly assembled tailocins. The two amino acid junctions further impact efficacy, at least for the evolutionary optimized native R2 tailocin of *P. aeruginosa*. Plausible causes are an inferior orientation of the RBP, a negatively impacted signal cascade linking cell binding and the eventual killing mechanism, or differences in RBP expression yield.

### 2.4. Construction of Nine Chimeric RBPs Targeting E. coli Serogroups O26, O103, O104, O111, O145, O146, and O157

The wild-type R2 tailocin was engineered by swapping the RBD of the RBP with those of various phages targeting *E. coli*. This engineering was performed at the DNA level using the VersaTile technique. The chimeric RBPs were subsequently expressed in trans, as for R2-WT-VT. Thus, the first step was to create chimeric RBPs using the anchor of the wild-type R2 tailocin and the RBDs of the different phage RBPs. The C-terminal RBDs targeting the *E. coli* O-antigen originate from phages with a podovirus morphology. This includes the RBPs from the Escherichia phage PAS7 (targeting serogroup O103), belonging to the recently proposed genus *Cepavirus* [35] and the Escherichia phage PAS61 (targeting serogroup O146) and O157 typing phage 10 (Tp10 for short; targeting serogroup O157), both belonging to the genus *Uetakevirus*. Other RBPs originate from prophages belonging to the *Lederberg* (targeting serogroup O111 and O145) and *Uetakevirus* (targeting serogroup O26) genera that were identified within the genomes of bacterial strains with the desired serogroups, namely *E. coli* strains RM10386 (serogroup O26), 110512 (serogroup O111), and FHI58 (serogroup O145). Fusions between the N-terminal R2 anchor and the C-terminal RBDs from the mentioned *E. coli* (pro)phages using VersaTile were made to retarget the bactericidal spectrum of the native R2 tailocin towards specific *E. coli* O-antigen serogroups. The amino acid sequences and AlphaFold3-predicted structures of all chimeric RBPs can be found in Appendix A, respectively.

These engineered R2 tailocins were produced and enriched using high-speed centrifugation, with protein yields ranging from 439 to 857 µg/mL (Appendix A). For two engineered R2 tailocins containing RBDs against serogroups O103 and O145, an alternative shuttle expression vector (pVTD27) was used for comparison. pVTD27 is identical to pVTD29 but does not contain a *lac* repressor (*lacI* gene). The *lac* repressor prevents leaky expression from the used *lac* promotor, which is induced by isopropyl ß-D-1-thiogalactopyranoside (IPTG), generally lowering the expression level of the RBP gene prior to induction.

### 2.5. All Engineered R2 Tailocins Targeting E. coli O-Antigens Are Bactericidal Against at Least One Susceptible Strain with Highly-Variable Efficacy

In a spot assay, zones of clearance were observed on at least one susceptible strain for all engineered R2 tailocins targeting a specific *E. coli* O-antigen (Figure 3, Appendix A). The best-performing engineered R2 tailocins are R2-O145-b, R2-O103-b, and R2-O157, and R2-O111, showing clearance at 0.07 µg/mL, 0.6 µg/mL, 0.6 µg/mL, and 2.3 µg/mL (corresponding to 3.5, 30, 30 and 115× higher concentrations than the average R2-WT). R2-O26, R2-O104, and R2-O146 were active, but remarkably higher doses (75–300 µg/mL) were needed to be visible as a spot. Some strains were insensitive to the corresponding engineered tailocins. These observations and the variability therein indicate that the R2 tailocin scaffold is remarkably amenable to accepting a diversity of new RBDs. It should be emphasized that the newly grafted RBDs in this study have a tailspike morphology in contrast to the native tail fiber structure. Nevertheless, the inferior activity compared to the native evolutionarily optimized R2 tailocin of *P. aeruginosa* shows that there is further technical optimization potential (delineation, shuttle vector, different RBD source), as seen for the two variants of R2-O103 and R2-O145. In addition, a horizontal transfer event in nature is accompanied by adaptive evolution, particularly surrounding the newly formed junction site, to optimize the functioning of the new chimera. It is plausible that mutagenesis around the junction site will further enhance the antibacterial effect.

A survival assay was performed for all R2 tailocin constructs targeting the *E. coli* O-antigen, as was performed for the R2 tailocin controls R2-WT, R2-WT-trans, and R2-WT-VT. R2-O145-b, R2-O111, and R2-O157 caused the highest reduction in bacterial cell count (between 3 and 4.6 log across the susceptible *E. coli* strains tested) in the survival assay, with 1.6, 3.1, and 3.1 µg/mL as the lowest concentration with significant reduction, respectively, followed by R2-O103-b (12.5 µg/mL; 3.4–3.7 log) (Table 1; Figure 4). Other engineered tailocins required doses higher than 150 µg/mL, and for R2-O26, no significant reduction in cell count could be observed.

In addition to the survival assay, a growth inhibition assay was performed for all engineered R2 tailocins. This growth inhibition assay was initiated from a low inoculum (±10^5^ CFU/mL). Growth was monitored over a period of 24 h upon addition of the engineered tailocin. Growth was evaluated at two different time points. (I) After 8 h, the difference in optical density between various estimated concentrations of the engineered R2 tailocin and the untreated sample was evaluated. Statistical analysis was performed using the untreated sample as a reference. However, in cases where a significant inhibiting effect was detected for the negative control containing R2Δ*prf15* (Control B), this control was used as reference instead. (II) After 24 h, the minimum inhibitory concentration (MIC) for full inhibition was investigated. 

The growth inhibition assay resulted in variable outcomes for the engineered R2 tailocins targeting the *E. coli* O-antigen. Growth inhibition evaluated at 8 h was the most sensitive metric to identify (partially) functional tailocins, with a significant impact on the growth curve at 8 h for all engineered tailocins targeting *E. coli* (Table 1; Figure 4; Appendix A). Generally, higher doses were needed to maintain a significant impact on the growth curve after 24 h, which can be explained by an incomplete eradication of the targeted bacteria or the growth of emerging resistant clones.

### 2.6. Construction of Five Chimeric RBPs Targeting E. coli Capsular Serotype K1 and K. pneumoniae Capsular Serotypes K11, K31, and K63

The tailocin engineering approach based on the VersaTile assembly technique allowed rapid screening of a large diversity of RBDs for the construction and identification of functionally engineered tailocins customized to an array of O-antigens. This standardized approach was then further applied to address another cell surface receptor and species. Many *E. coli* strains also possess a polysaccharide capsule (K-antigen) to increase their pathogenicity, with K1 as the most prevalent capsule serotype [36]. Therefore, to create chimeric RBPs, we selected the RBD of the K1 capsule-specific Escherichia phage K1F (K1Fgp17, podovirus morphology, genus *Kayfunavirus*). In addition, we sourced RBDs from several *Klebsiella* phages targeting capsular serotypes to create *Klebsiella*-specific R2 tailocins. Specifically, we selected RBDs from phages with podovirus (phages K11 and KP34) and siphovirus (phage KP36) morphologies. Both phages KP34 (*Drulisvirus*) and KP36 (*Webervirus*) possess a single RBD with 42.4% aa similarity (94% coverage) between them, recognizing the same capsular serotype K63, designated here as RBD-K63-a and RBD-K63-b, respectively. Phage K11 belongs to the genus *Przondovirus* and harbors a dual-branched RBP system, with each RBP having a different receptor specificity (capsular serotype K11 and K31) [37]. All selected RBDs were fused to the N-terminal R2 anchor, and engineered tailocins were produced and enriched by high-speed centrifugation, resulting in protein concentrations ranging from 273 to 470 µg/mL (Appendix A).

### 2.7. Engineered Tailocins Can Target Capsule as Receptor Against Both E. coli and K. pneumoniae

R2-K1 was spotted against the panel of *E. coli* and *K. pneumoniae* strains (Figure 5, Appendix A). Besides capsule K1 producing *E. coli* strains CUGG28 and CAB1, R2-K1 also showed clearance on the *E. coli* strain ECOR28 of serogroup O104, of which the capsular serotype was unknown. A blastn analysis was performed to search for similarities between the ECOR28 strain genome (GCF_002190675.1) with serotype-specific capsule biosynthesis proteins [36]. This analysis revealed similarities (>97% coverage) to the proteins encoded by genes *neuD*, *neuB*, *neuA*, and *neuC* (respective UniProtKB accessions: Q46674, Q46675, A0A0D6H548, and Q47400) responsible for K1 capsule biosynthesis, with 59, 69, 48, and 59% aa identity to their respective protein sequences. However, no similarity was found to the remaining K1 capsule biosynthesis proteins encoded by genes *neuE* and *neuS* (UniProtKB accessions: Q47401 and Q47404). The *Klebsiella* strains were not susceptible to tailocin R2-K1.

All *Klebsiella*-targeting tailocins were spotted against a panel of *K. pneumoniae* strains with different capsular serotypes. The observed susceptibility fully agrees with the host specificity of the corresponding phage sources (Figure 5). R2-K63-b with the RBD of Klebsiella phage KP36 with a siphovirus morphology demonstrated the most effective clearance, starting from an estimated R2 tailocin concentration of 25 µg/mL. In contrast, R2-K63-a derived from podophage KP34 required a 12-fold higher dose. These observed differences demonstrate the value of an approach to rapidly construct and screen a diversity of RBDs in a standardized manner. Additionally, the RBD of K11gp43, targeting capsular serotype K31 was grafted into the R2 tailocin scaffold (R2-K31). Unfortunately, no spots were observed for R2-K31. However, only one *K. pneumoniae* strain with capsular serotype K31 was tested which could not be infected by phage K11. Therefore, this line of investigation was discontinued.

Survival and growth inhibition assays were conducted for tailocins that exhibited activity in the spot assay (Figure 6; Table 1, Appendix A). R2-K1 and R2-K11 showed a significant bactericidal effect against their susceptible host, starting from an estimated R2 tailocin concentration of 25 and 50 µg/mL, respectively. At an estimated R2 tailocin concentration of 100 µg/mL, a bactericidal effect of 1.6 and 1.1 log (CFU/mL) reduction in bacterial population was observed for R2-K1 and R2-K11, respectively. Additionally, all four capsule-targeting R2 tailocins showed a significant impact on the growth curve at 8 h. These observations emphasize again that this assay most easily captures functional tailocins, even when having inferior activity necessitating further optimization.

It must be noted that the R2 tailocin concentration at which a bactericidal effect is observed is dependent on the enrichment method. Tailocin R2-K11 was produced using two different methods (Appendix A). This tailocin showed a significant bactericidal effect in the survival assay from an estimated concentration of 50 µg/mL after enrichment by high-speed centrifugation and from 6.2 µg/mL after enrichment by AS precipitation. Similarly, at a R2 tailocin concentration of 100 µg/mL in the survival assay, an increase in activity from 1.1 ± 0.4 up to 6.3 ± 1.3 log (CFU/mL) was observed for R2-K11 when AS precipitation was the chosen enrichment method. These observations indicate that the purity and/or intactness of the tailocins is higher in the case of AS precipitation. However, high-speed centrifugation excels in convenience.

### 2.8. Construction of a Bivalent R2 Tailocin R2-K11-K1 with Activity Against Both K. pneumoniae Capsular Serotype K11 and E. coli Capsular Serotype K1

The VersaTile DNA assembly method is dedicated to the rapid combinatorial construction of modular proteins and allows the construction and evaluation of more advanced RBP architectures in engineered tailocins to be pursued. As explained for Klebsiella phage K11, many phages have a dual RBP architecture, with the first RBP attached via its anchor domain to the phage tail. A second RBP then docks via its N-terminal conserved peptide (CP) on a branching domain present in the first RBP, which is located directly after the anchor domain. Specifically in phage K11, the first RBP of phage K11 (K11gp17) with K11 specificity (RBD-K11) is connected to the second RBP of phage K11 (K11gp43) with K31 specificity via such a branching domain–CP connection.

To transplant such a dual RBP architecture into the R2 tailocin scaffold, the RBP cluster was built by assembling four tiles sourced from different tailocins or phages: (I) the anchor tile sourced from the native R2 tailocin; (II) RBD-K11 of Klebsiella phage K11 targeting the *Klebsiella* capsular serotype K11, which includes the T4gp10-like branching domain that serves as a crucial docking site for the second RBP; (III) the CP tile from the second RBP, KP32gp38, of the Klebsiella phage KP32 [38], which is highly similar to its equivalent in K11 [37]; and (IV) RBD-K1 from Escherichia phage K1F targeting the *E. coli* capsular serotype K1 (Figure 7A,B).

This VersaTile-assembled fragment encoding the dual RBP cluster was subsequently expressed in trans as described above. Similarly to R2-K11 and R2-K1, which were produced simultaneously with R2-K11-K1, the R2-K11-K1 tailocin was purified by high-speed centrifugation and spotted onto the panel of *E. coli* and *K. pneumoniae* strains (Figure 5). R2-K11 showed clearance on the *K. pneumoniae* strain 390 (K11 capsule), while R2-K1 showed clearance on *E. coli* strains CUGG28 (capsular serotype K1), CAB1 (capsular serotype K1) and ECOR28 (serogroup O104, capsular serotype unknown), with no cross-activity observed. In contrast, R2-K11-K1 displayed a broadened target range, targeting both capsular serotypes, visible as opaque spots. The bactericidal activity of the bivalent tailocin was further evaluated against its susceptible hosts, namely *K. pneumoniae* strain 390 (capsular serotype K11) and *E. coli* strain CUGG28 (capsular serotype K1), using the survival assay. A significant bactericidal effect was observed against both strains, starting from 50 µg/mL (Figure 7C). The activity of R2-K11-K1 at an estimated R2 tailocin concentration of 100 µg/mL in the survival assay was similar to the activity of R2-K1 and R2-K11, with a 1.4–2.0 log (CFU/mL) reduction in bacterial population (Table 1). A significant impact on the growth curve after 8 h could be observed from 0.4 µg/mL and 25 µg/mL for the *K. pneumoniae* capsular serotype K11 and *E. coli* capsular serotype K1, respectively (Figure 7D). A significant MIC after 24 h was observed for *K. pneumoniae* strain 390 (12.5 µg/mL) and *E. coli* strain CCUG28 (50 µg/mL). The impact on the growth curve provoked by the second RBP, targeting *E. coli* K1 capsule, was thus less distinct than for the first RBP, targeting *K. pneumoniae* K11 capsule.

## 3. Discussion

In this work, we have converted the tailocin engineering and production method described previously [23] into a plug-and-play framework that facilitates the rapid and straightforward integration of RBPs from diverse origins. We have evaluated this tailocin platform with diverse RBPs targeting the O-antigen or capsule (K-antigen) of *E. coli* and *K. pneumoniae*. In addition, we have also demonstrated that a branched RBP system targeting capsular serotypes of two different species can be transplanted into the R2 tailocin scaffold. In total, eight tailocins were redirected towards the O-antigen of *E. coli*, one towards the *E. coli* capsule and three towards the *K. pneumoniae* capsule, with one bivalent R2 tailocin targeting both the capsule of *K. pneumoniae* and *E. coli.* The main progress of this work compared to state-of-the-art tailocin engineering is the scale (i.e., the high number of engineered tailocins in a single study), the development of the first tailocins redirected towards the bacterial capsule and *K. pneumoniae,* and the first application of a branched RBP system in a tailocin scaffold.

### 3.1. VersaTile Applications to Engineer Tailocins

The enhancement in scale was achieved through the introduction of VersaTile (a DNA assembly technique dedicated to the combinatorial engineering of modular proteins such as RBPs) in the tailocin engineering process. In particular, the branched RBP system excels in modularity and is a prime example of how VersaTile can facilitate the rapid construction and exploration of modular RBP systems in a tailocin scaffold. This study expands on earlier work where the VersaTile technique was successfully implemented for other modular proteins such as phage lysins [29] and innolysins [39]. In addition, the VersaTile technique was successfully applied to create chimeric tailspikes by recombining the N-terminal anchor and C-terminal RBDs of four different *Klebsiella* phages [21]. The efficiency of this assembly process can be attributed to the use of a single type IIs restriction enzyme that allows simultaneous restriction–ligation in contrast to traditional cumbersome cloning techniques that were used previously for tailocin engineering. The VersaTile junction affected the antibacterial activity of R2 tailocins. However, traditional cloning methods also result in junctions without the flexibility to choose the amino acids, and junction-less assembly lacks scalability because of the custom design process per engineered tailocin or synthesis cost (~150 EUR/construct). Upon the identification of functionally engineered tailocins, these techniques can be used to further tune hit-to-lead development and improve biological activity.

### 3.2. R2 Tailocin as the Scaffold for Enzymatic Tailspikes and Extra-Species RBPs

An array of engineered R2 tailocins with a chimeric RBP were successfully created targeting the *E. coli* O-antigen serogroups O26, O103, O104, O111, O145, O146, and O157, the *E. coli* capsular serotype K1, and *K. pneumoniae* capsular serotypes K11 and K63. The majority of the newly engineered R2 tailocins, including R2-O104 and R2-O157, R2-O103-b, R2-O111, R2-O145-b, R2-O146, R2-K1, and R2-K11, showed clearance in the spot assay, bactericidal activity in the survival assay, and a significant impact on the growth curve in the growth inhibition assay against their susceptible host(s). We proved that the *P. aeruginosa* R2 tailocin can be used as a scaffold to integrate extra-species RBPs with O-antigen and capsule specificity, showing activity against the respective serotype/host. Notably, the RBPs used in this work have a tailspike morphology featured by a β-helix in contrast to the fibrous tail fiber morphology of the native R2 RBP. Correct delineation of the RBD and optimization of the junction site are major aspects impacting the optimal functioning of newly engineered tailocins and should be the initial focus for further tailoring. In this work, we also predicted the quaternary structure of the chimeric RBPs using AlphaFold3, with good confidence (pLDDT > 70) for the majority of amino acid positions within the proteins (Appendix A). However, no correlations could be observed between the quaternary structure of the chimeric RBPs and the efficacy of the engineered R2 tailocins. Structure predictions thus facilitate a functional delineation of the RBD, but in the current stage, empirical testing of a few delineations remains relevant.

### 3.3. R2 Tailocin as the Scaffold for Complex RBP Systems Enabling Cross-Species Activity

The first bivalent R2 tailocin R2-K11-K1 with activity against both the capsular serotype K11 of *K. pneumoniae* and capsular serotype K1 of *E. coli* emphasized the surprising engineering potential of the tailocin scaffold, here facilitated by the flexibility of VersaTile and the availability of a diverse repository of tiles of RBP modules. The chimeric RBP cluster with bivalent host specificity was created by reusing tiles from this in-house existing tile repository, which contains non-homologous, modular RBP building blocks sourced from different *Escherichia* and *Klebsiella* phages (Figure 7). Latka et al. already demonstrated that the anchor domain of Klebsiella phage KP32 could be fused with the enzymatic domain of RBP1 of Klebsiella phage K11 and successfully integrated this chimeric RBP in the phage K11 scaffold [21]. This replacement was feasible as both phages belong to the same taxonomic genus and have a high identity of 88% at the amino acid level between their anchor domains [21,37]. Combining these findings with other unpublished data, an inter-species branched RBP cluster was created consisting of the RBD of Klebsiella phage K11gp17 with K11 capsule specificity in the first position and the RBD of Escherichia phage K1Fgp17 with K1 capsule specificity in the second position. The anchor-branched system relied on the T4gp10-like domain present in the Klebsiella phage K11gp17 and the CP derived from the Klebsiella phage KP32gp38. Subsequently, the cluster was attached to the R2 tailocin scaffold using the native R2 anchor domain. In total, building blocks derived from four different origins (*Pseudomonas*, *Escherichia*, and two different *Klebsiella* phages) were used to create the first cross-species bivalent tailocin R2-K11-K1.

### 3.4. Technical Aspects of Engineered Tailocin Production Impacting the Final Antibacterial Effectiveness

The number of R2 tailocin particles in the survival assay can be estimated with a formula based on the Poisson distribution [23]. This number is relevant to calculate the tailocin-to-cell ratio, as tailocins are single-hit molecules and multiple tailocin particles may be randomly attached to a single bacterial cell, or conversely, some bacterial cells may not encounter any tailocin at all [40]. However, the efficacy of the engineered R2 tailocins on each strain varies, meaning that this estimate of the number of killing particles would become strain dependent. We therefore chose to assess the R2 tailocin efficacy based on protein concentration, as performed for traditional antimicrobial assays. This approach was also applied by Redero and colleagues when testing the activity of wild-type R-type tailocins in a murine pneumonia model [41] and by Latrova and colleagues to characterize the pore-forming activity of R-type tailocins targeting *Pragia fontium* [42]. A disadvantage of this method is that the R2 tailocin is not highly pure after a single enrichment step, resulting in an overestimate of the tailocin particle number [42].

R-type tailocin enrichment by AS precipitation [23,28,40,42,43], PEG precipitation [41,44,45,46], ultra-centrifugal filter units [28], and/or high-speed centrifugation [23,24,26,28,40,41,42,43,45,46] and combinations thereof have been reported. In the course of this study, we switched from AS precipitation to high-speed centrifugation due to its faster processing time and its higher amenability for upscaling, although the latter method resulted in a 8-fold lower bactericidal activity in the survival assay for R2-K11 (Appendix A). Thus, R2 tailocin enrichment using high-speed centrifugation or AS purification alone can provide an initial indication of the tailocin activity, but a higher tailocin purity is needed to assess the activity in greater detail. Combining these enrichment methods with additional purification methods, such as chromatographic affinity-based methods [47,48,49,50], gel filtration [6,50,51], or density gradient centrifugation [52,53], can increase the tailocin purity. Moreover, the protein concentration does not inform on the intactness and completeness of tailocin particles. An imbalance of the produced chimeric RBPs in relation to the available R2 scaffolds may lead to dysfunctional, partially assembled, or misassembled particles. Mass spectrometry could be applied to selected engineered tailocins to gain a deeper understanding of the complete assembly and the assembly kinetics [54]. In addition, it must be noted that F-type tailocins produced under the same *recA* regulator are possibly still present in the engineered R2 tailocin preparation. The removal of unnecessary genes under the *recA* regulator could potentially improve the activity determination of the tailocin product. Generally, a higher purity and characterization of the tailocin complex leads to a better quantification and thus to a better comparison of the engineered tailocins, but at the expense of the screening scale. Therefore, as a first exploratory step, we chose a high-throughput diversification approach in combination with a screening of less-pure tailocin preparations instead of highly purified tailocin preparations for reasons of efficiency.

To easily detect newly engineered, functional tailocins, we found that an assessment of the growth curve after 8 h in the growth inhibition assay proved to be the most suitable method, with a statistically significant impact for all engineered tailocins. Interestingly, a statistically significant impact on the growth curve of R2-O157 was observed on the target *E. coli* O157 strain 264, although there were no clear zones visible when spotting on bacterial lawns, underlining the relevance of the growth inhibition assay for initial testing, even if the spot assay is negative. However, whereas R2-O26, R2-K63-a, and R2-K63-b showed a significant impact on the growth curve, no bactericidal effect could be observed under the tested conditions. These observations confirm that the growth inhibition assay is more sensitive than other assays, which can be attributed to the higher tailocin-to-target cell ratio. Although R2Δ*prf15* did not show clearance in the spot assay on the respective host of the engineered tailocins, this control (control B) showed a significant effect on the growth curve in the growth inhibition assay for R2-O26, R2-O145-b, R2-O146, and R2-K11, complicating the interpretation of the results of these tailocins. As it cannot be ruled out that the observed growth curve effects were caused by the F-type tailocins present in control B, this control was used as the reference for statistical analysis instead of the untreated sample. However, it must be noted that R2Δ*prf15* was only tested at the highest concentration, unlike the engineered tailocins which were tested at a range of diluted concentrations. This in turn may have led to an underestimation of the inhibitory effect of the engineered R2 tailocins.

Interestingly, the switch from shuttle expression vector pVTD29 to pVTD27 for the two tailocin constructs R2-O103-a and O145-a to obtain R2-O103-b and R2-O145-b, respectively, increased the antibacterial activity of the engineered tailocins. When shuttle expression vector pVTD27 without *lac* repressor was used to create R2-O103-b and R2-O145-b, a 16- and 540-fold improvement could be observed in the spot assay (Figure 3). Additionally, these -b variants performed 20 and 125 times better than the -a variants in the survival assay and 1260 times better in the growth inhibition assay at 8 h for R2-O103 and O145, respectively (Table 1; Figure 4). These results indicate that an increased in trans expression and possible accumulation of RBPs prior to induction of the R2 tailocin scaffold (due to the absence of the *lac* repressor) can significantly improve the quantity and proportion of completely assembled R2 tailocins. Thus, the amount of RBP may be the limiting factor for the assembly of intact tailocins. This increased RBP production may lead to more correctly or completely assembled tailocin particles, consequently resulting in lower required doses to exert a significant killing or growth-inhibitory effect. To further investigate this hypothesis, the R2 tailocins should be further purified to allow detailed comparisons across the results. Additionally, the utilization of a shuttle expression vector featuring a tunable promotor capable of regulating gene expression levels (e.g., an arabinose-inducible promotor) could be explored. Alternatively, a PCR amplicon or synthetically ordered fragment of the engineered RBP can be provided in vitro into a cell-free system to produce tailocins [55].

### 3.5. Tailocins, RBPs, and Phages as Therapeutic and Diagnostic Tools

Tailocins and phages both have antibacterial properties but differ in their mode of action. Tailocins are non-replicative proteins that rely on a single-hit mechanism, binding to and puncturing the cell envelope to kill the cell. In contrast, phages are replicative entities that, after receptor recognition, inject their genetic material into the host, initiating a replication cycle that ultimately leads to bacterial lysis and cell death. Phages have a so-called auto-dosing effect of continued replication as long as new hosts can be infected. Yet, replication is also associated with possible evolutionary changes in the replicated phage genome. Enzymatic RBPs, on the other hand, only have antivirulence activity and do not directly kill bacterial cells, but actively degrade surface polysaccharides, making the bacterium again more susceptible to immune system clearance of other conventional therapeutics [56]. Tailocins and RBPs have more resemblance to conventional pharmaceuticals, thereby circumventing regulatory, ethical, and legislative difficulties related to the use of phages.

One important factor to consider is the difference in activity range between phages and tailocins. Two observations can be made on the difference between the efficacy of R2 tailocins and phages in this work. First, the Escherichia phage Tp10 containing the RBD that was used to create the chimeric RBP in R2-O157 only infects one out of six *E. coli* strains of the respective serogroup, namely the *E. coli* strain Sakai [57]. However, R2-O157 kills five out of six *E. coli* strains (332, 584, 777/1, 2905 and Sakai), with an additional impact on the growth curve for *E. coli* strain 264 (Figure 4). This result of R2-O157 suggests a broader activity range of R2 tailocins compared to phages. This can be explained by the presence of intracellular anti-viral defenses restricting the host spectrum of phages but not of tailocins. Secondly, the Escherichia phage PAS7, delivering the RBD for tailocin R2-O103, infects the *E. coli* strain 4215/4 and causes lysis from without on *E. coli* strains NVH-848 and P11-2315. This lytic activity on all three *E. coli* O103 strains suggests that the Escherichia phage PAS7 can successfully bind to its host [58]. In contrast, the engineered tailocin R2-O103 kills *E. coli* strains 4215/4 and NVH-848, but not P11-2315. Additionally, some other strains in this work are resistant to R2 tailocins, although having the predicted serogroup/capsular serotype. This variable susceptibility of different strains towards a single tailocin may be explained by other confounding factors such as possible masking of the O-antigen.

Based on the high RBP specificity, the engineered R2 tailocins could potentially also be applied as diagnostics for bacterial typing instead of phages, avoiding the bias introduced by phage defense systems. Tailocins may be superior to RBPs for diagnostic purposes since the tailocin-induced lysis effect may be easier to detect compared to the halo zone produced by enzymatic RBPs.

## 4. Materials and Methods

**Media, bacterial strains, plasmids, and bacteriophages.** Appendix A lists all bacterial strains and phages used in this study, including their source. *K. pneumoniae* and *P. aeruginosa* strains were cultured in tryptic soy broth (TSB; Oxoid, Basingstoke, UK) and on tryptic soy agar (TSA; MP Biomedicals, Solon, OH, USA) at 37 °C. *E. coli* strains were grown in lysogeny broth (LB; 1% (*w*/*v*) tryptone, VWR, Radnor, PA, USA; 0.5% (*w*/*v*) yeast extract, VWR; 1% (*w*/*v*) NaCl, Fisher Scientific, Thermo Fisher Scientific, Waltham, MA, USA) and on LB agar (LA; LB supplemented with 1.5% (*w*/*v*) bacteriological agar, VWR) at 37 °C. Soft agar used in spotting assays contained 0.5% (*w*/*v*) agar (VWR). Each tile from the tile repository (Figure 1) was initially cloned and stored in vector pVTE (Appendix A), as previously described [29]. *E. coli* TOP10 cells (Invitrogen, Carlsbad, CA, USA) transformed with cloning vector pVTE were grown on LB supplemented with 100 μg/mL ampicillin (Fisher Scientific, Thermo Fisher Scientific, Waltham, MA) and 5% (*w*/*v*) sucrose (Fisher Scientific). For the assembly and storage of the engineered tailocin RBP construct, pVTD27 or pVTD29, two modified versions of the pUCPtac expression vector [23] were created (Appendix A). A *sacB* coding sequence, flanked with two BsaI recognition sites and two distinct six-nucleotide-long position tags (start (P_start_) and end (P_end_)) into which the BsaI restriction sites were embedded, was inserted downstream of the *tac* promotor. As a result, the *sacB* gene along with the BsaI recognition sites are removed during the VersaTile assembly reaction and replaced by the assembled RBP construct. Gentamicin (Carl Roth, Karlsruhe, Germany) and sucrose (Fisher Scientific) at final concentrations of 20 μg/mL and 5% (*w*/*v*), respectively, were supplemented to the media for *E. coli* TOP10 (Invitrogen) or *P. aeruginosa* to be transformed with expression vector pVTD27 or pVTD29 (Appendix A).

**Engineering of the RBP gene.** To create chimeric RBPs, anchor and RBD nucleotide sequence tiles were constructed. The anchor tile, derived from the native R2 tailocin, was created to facilitate the integration of different C-terminal RBD tiles from *E. coli* and *K. pneumoniae* phages in the R2 tailocin scaffold. Fragments of interest were amplified with Phusion High-Fidelity DNA Polymerase (Thermo Fisher Scientific) using VersaTile-specific primers featuring extended 5′ sequences to create the different tiles (Appendix A). The subsequent amplicons were verified with gel electrophoresis on a 1.2% (*w*/*v*) agarose (Chem-Lab, Zedelgem, Belgium) gel and purified with the GeneJET Gel Extraction Kit (Thermo Fisher Scientific Baltics UAB, Vilnius, Lithuania). These amplicons were inserted in the pVTE vector by restriction–ligation (SapI; Thermo Fisher Scientific; T4 DNA ligase; Life Technologies, Carlsbad, CA, USA) (Appendix A). Subsequently, the anchor and RBD tiles were fused in the shuttle expression vector pVTD29 in the VersaTile assembly reaction (Appendix A). Two tailocin variants were created in both pVTD27 and pVTD29, which are identical, but the former lacks the *lac* repressor (*lacI* gene). Cloning in pVTD27 results in the presence of a nine instead of six-nucleotide junction between the anchor and the RBD (encoding Leu-Gly-Ser vs. Gly-Ser) resulting from the original vector backbone (pM50) [39] and the assembly junction site. All transformations in the VersaTile cloning and assembly steps are performed using chemically competent *E. coli* TOP10 cells (Invitrogen). Plasmid extraction was performed using the GeneJet plasmid miniprep Kit (Thermo Fisher Scientific Baltics UAB). Overview figures were created using Affinity Designer 2 version 2.4.2.

**Quaternary structure prediction of the chimeric RBPs.** The trimeric proteins were predicted using AlphaFold3 [59] on the online webserver (https://alphafoldserver.com, accessed on 12 December 2024). The reliability of the AlphaFold predictions was assessed by the Local Distance Difference Test (LDDT) score reported for each structure (Appendix A). Structures were further processed using PyMol version 2.5.2. Figures were further optimized using Adobe Illustrator version 26.5.1 and Adobe InDesign version 16.4.3.

**R2 tailocin production and purification.** R2 tailocin expression and purification were based on previous research that was performed [23]. Briefly, 18 h cultures of *P. aeruginosa* PAO1 Δ*prf15* strains containing the pVTD27 or pVTD29 expression vector (depending on the construct, Appendix A) containing the engineered RBP coding sequence in trans, were grown and 1:100 diluted in 15 mL G medium (20 g/L monosodium L-glutamic acid) (Sigma, St. Louis, MO, USA), 5 g/L glucose (Carl Roth), 2.67 g/L Na_2_HPO_4_·2H_2_O (Fisher Bioreagents, Thermo Fisher Scientific), 100 mg/L MgSO_4_·7H_2_O (VWR), 250 mg/L KH_2_PO_4_ (Carl Roth), and 500 mg yeast extract ((VWR); pH 7.2) supplemented with 20 µg/mL gentamicin (Carl Roth) and incubated at 37 °C while shaking at 225 rpm. At an OD_600_ of 0.25, 3 µg/mL mitomycin C (Sigma-Aldrich St. louis, MO, USA) was added to induce the tailocin genes in the *P. aeruginosa* PAO1 Δ*prf15* genome. When a RBP was expressed in trans, the cultures were additionally supplemented with 0.25 mM isopropyl β-D-1-thiogalactopyranoside (IPTG; Carl Roth) to induce the *tac* promotor. Thereafter, the culture was incubated (37 °C, 180 rpm) for 2.5 h until complete lysis was observed. Subsequently, 0.1 U/mL DNaseI (Thermo Fisher Scientific) was added, and the culture was additionally incubated for 30 min. After incubation, the cultures were centrifuged (Eppendorf 5810R; Eppendorf, Hamburg, Germany) at 20,000× *g* for 1 h at 4 °C to remove the bacterial debris. Two methods of tailocin purification were performed (Appendix A). (I) A 4 M ammonium sulfate (AS, Carl Roth) solution was dropped at a slow flow rate of 1 mL/min into the supernatant while shaken on ice until a final concentration of 1.6 M was reached. The resulting suspensions from each replicate were thereafter stored for 18 h at 4 °C while shaken at 160 rpm. The AS concentrate was precipitated by centrifugation (Eppendorf 5810R) at 20,000× *g* for 1 h at 4 °C and resuspended in 10% (*v*/*v*) of the start volume with cold TN50 buffer (1.21 g/L Tris-HCl (VWR), 2.92 g/L NaCl (Thermo Fisher Scientific); pH 7.5). (II) Alternative to AS precipitation, tailocins were precipitated using a high-speed centrifuge (Sorvall Lynx 6000, Thermo Fisher Scientific) at 80,000× *g* for 1 h at 4 °C and resuspended in 10% (*v*/*v*) of the start volume with cold TN50 buffer [24]. Each tailocin solution was filter sterilized with a 0.45 µm PES syringe filter (Novolab, Geraardsbergen, Belgium) to remove residual impurities. To evaluate the production host, the following three controls were included: (I) native R2 tailocin produced in *P. aeruginosa* PAO1 strain (R2-WT), (II) *P. aeruginosa* PAO1 Δ*prf15* strain containing the pVTD29 expression vector encoding the native *prf15/16* gene in trans (R2-WT-trans), and (III) *P. aeruginosa* PAO1 Δ*prf15* strain containing the pVTD29 expression vector encoding the VersaTile-assembled *prf15/16* coding sequence in trans (R2-WT-VT). All expressions were performed in triplicate. An estimation of the tailocin concentration was made using the Micro BCA^TM^ Protein Assay Kit (Thermo Fisher Scientific) following the manufacturer’s instructions.

**Determination of the specificity spectrum.** After tailocin expression, the specificity spectrum of the engineered R2 tailocin was determined by two spot assays. First, for each of the three replicates of engineered R2 tailocin production, an 18 h culture was grown for the four *K. pneumoniae* strains or/and twenty *E. coli* strains, and *P. aeruginosa* wtb/CF510, in triplicate. Next, a bacterial lawn was prepared from each culture using the soft agar method [60]. This was achieved by adding 200 µL of the culture to 4 mL of molten soft agar that was subsequently poured onto a TSA/LA plate and left to dry. A 5 µL drop of the highest tailocin concentration was spotted on all bacterial lawns. Plates were then incubated for 18 h at 37 °C. Bacterial susceptibility became visible by the appearance of a clear/opaque, circular zone on the spot site. For comparison purposes, phages present in our collection, specifically Escherichia phages PAS7 (GenBank accession: OQ921331.1), PAS61 (GenBank accession: OQ921333.1), O157 typing phage 10 (Tp10; GenBank accession: KP869108.1), K1F (GenBank accession: NC_007636.1), Klebsiella phages K11 (GenBank accession: NC_011043), KP34 (GenBank accession: NC_013649.2), and KP36 (GenBank accession: NC_029099.1), from which the RBDs were transferred to the R2 tailocin scaffold, were also spotted on the bacterial strains containing the targeted receptor.

Once the susceptible host(s) were identified, an additional two-fold serial dilution of tailocins in TN50 buffer was made, starting from a concentration of 300 µg/mL. Next, 3 µL of each dilution was spotted on a bacterial lawn of the tailocin-sensitive host prepared according to the soft agar method and incubated for 18 h at 37 °C. The minimal concentration was determined, for which the tailocin gave an opaque spot. The results of this two-fold serial dilution spotting assay are shown in Appendix A. The TN50 buffer, the native R2 tailocin, and the RBP-deficient derivative (R2Δ*prf15)*, also termed the tailocin scaffold, were spotted as negative controls.

**Survival assay to quantify the bactericidal killing effect.** The culture of the target bacterial strain was grown for 18 h, diluted (1:100) in TSB/LB, and incubated at 37 °C (225 rpm) until the mid-exponential phase (OD_600_ = 0.6–0.7) was reached. Next, the bacteria were harvested by centrifugation (4000× *g*, 10 min) and resuspended in 1:1 volume TN50 buffer, followed by an additional washing step. The following dilution series was made in TN50 buffer for the expressed tailocin: 150, 100, 50, 25, 12.5, 6.25, 3.13, 1.56, 0.78, 0.39, 0.19, 0. 0.097, 0.046, and 0 µg/mL. To start the assay, a volume of 50 µL of each tailocin dilution was added to 50 µL of target bacteria. After 40 min of incubation at 37 °C, a ten-fold serial dilution of bacteria-tailocin mixture was made and 5 µL of the dilution series was spotted on a TSA/LA plate. After 18 h incubation at 37 °C, the colonies were counted and expressed in log_10_ scale (CFU/mL) and compared to the untreated control as an indication of tailocin bactericidal activity. The survival assay was performed for each of the three biological replicates. To test which tailocin treatments were significant across various concentration levels, a linear mixed model with random intercept per biological replicate was fitted to the data. Tailocin concentrations were considered as a categorical predictor. For each concentration, the outcome was compared to the untreated sample (tailocin concentration zero), while automatically adjusting for multiple testing using Dunnett’s approach. The impact of the different tailocin concentrations on the bacterial log-reduction compared to the untreated sample, termed the contrast, was plotted in relation to the tailocin concentrations with joint confidence intervals (Appendix A) [61]. The *p*-values were calculated to indicate the statistically significant differences in the results. Statistical analysis and data processing were performed in R Statistical Software (R 4.3.1). Figures were further processed using Adobe Illustrator version 25.4.1 and Adobe InDesign version 16.4.3.

**Growth inhibition assay to assess the inhibitory effect of an engineered R2 tailocin against its sensitive bacterial strain(s).** Varying concentrations (250, 200, 150, 100, 50, 25, 12.5, 6.25, 3.13, 1.56, 0.78, 0.39, 0.19, 0.097, 0.046, and 0.0046 µg/mL) of the R2 tailocin were tested. For each independently produced tailocin, a dilution (1:100) of an 18 h incubated culture was prepared and grown until reaching an OD_600_ of 0.08–0.1. Subsequently, a dilution (1:100) in 2× TSB/LB was prepared, corresponding to a final bacteria density of ~10^6^ CFU/mL. For each tailocin concentration, 50 µL of the bacterial suspension was added to 50 µL of the purified R2 tailocin derivative at the concentrations listed above. The optical density of the suspension was monitored at 600 nm using an Infinite 200 PRO^®^ reader (Tecan, Männedorf, Switzerland) at 15 min intervals over a 24 h duration at 37 °C. The complete 24 h monitoring period was visualized using R Statistical Software (v4.3.1, (Team, 2021)) and further processed using Adobe Illustrator version 25.4.1 and Adobe InDesign version 16.4.3. Negative controls consisted of (I) 50 µL of 2× TSB/LB added to 50 µL TN50 buffer (Blank), (II) 50 µL of 2× TSB/LB added to 50 µL R2 tailocin (highest available concentration) (control A), (III) 50 µL of the tailocin scaffold (R2Δ*prf15*) in the same concentration as control A added to 50 µL of bacterial suspension (control B), and (IV) 50 µL TN50 buffer added to 50 µL of bacterial suspension (concentration = 0 µg/mL).

The primary objective was to determine the concentration at which the tailocin completely inhibits bacterial growth. To achieve this, two different metrics were assessed based on the optical density measurements. As a first metric, a time point of approximately 8 h after exposure was chosen for all experiments based on the most substantial difference in the inhibition of bacterial growth between the concentrations of the engineered R2 tailocin and the untreated sample. At this selected time point, the means of the relative optical densities were calculated for each concentration of tailocin tested and plotted against their corresponding tailocin concentrations. The relative OD_600_ represents a measure of bacterial growth, obtained by subtracting the baseline value (Blank) from all OD_600_ readings and then normalizing each value against the OD_600_ value of the untreated sample (tailocin concentration zero), ensuring that all values are within the range of 0 to 1. Statistics were obtained using a Kruskal–Wallis test after failing assumptions, followed by a one-sided Dunnett’s test with the untreated sample as a reference. The resulting *p*-values were used to assess the significant differences with the untreated sample and were displayed in the figure presenting the results. When control B showed a significant inhibitory effect (for R2-O26 on strain ECA112162, R2-O145-b, R2-O146 and R2-K11), this control was used as a reference for statistics instead of the untreated sample to take any effects of the bacterial lysate into account. Note that the inhibition effect of the R2 tailocin could be underestimated, as control B was only tested at the highest protein concentration and no dilutions were tested. All graphics were created using R Statistical Software (R 4.3.1) and further processed using Adobe Illustrator version 25.4.1 and Adobe InDesign version 16.4.3. The second metric is the minimum inhibitory concentration (MIC), which corresponds to the lowest concentration at which full inhibition of growth was observed after 24 h. More specifically, full inhibition was assumed for the lowest concentration of R2 tailocin that inhibits ≥90% of the bacterial growth of the untreated sample at time point 24 h [62].

## Figures and Tables

**Figure 1 antibiotics-14-00104-f001:**
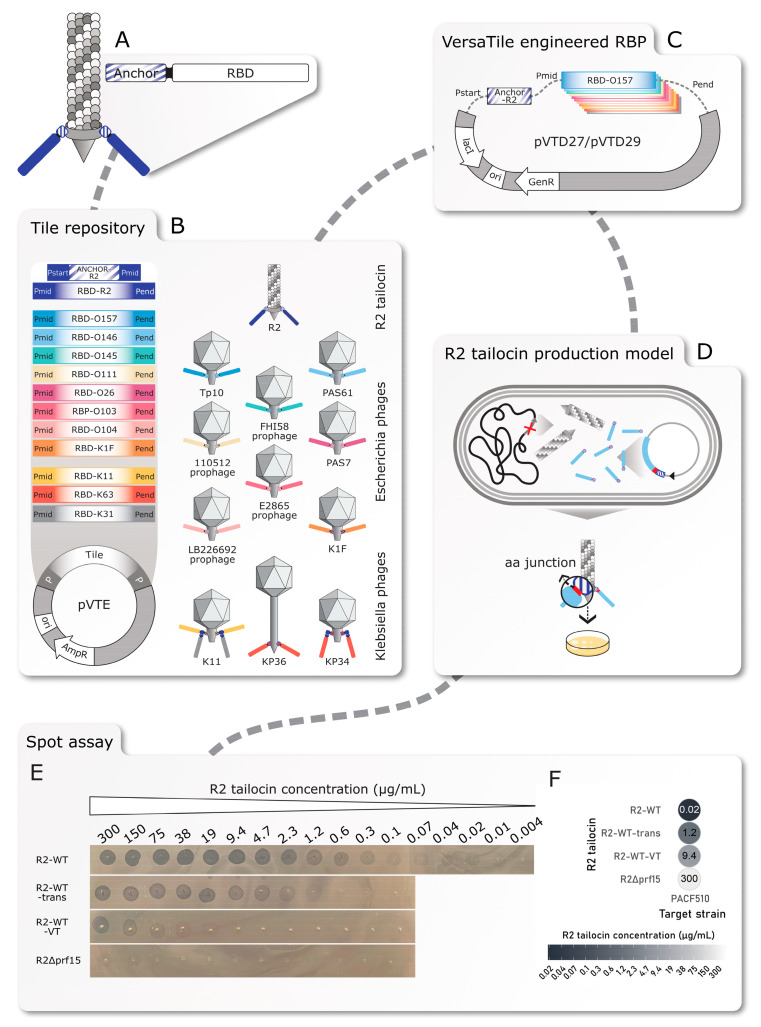
Pipeline for tailocin engineering and production. (**A**) Modular build-up of the R2 tailocin receptor-binding protein (RBP). (**B**) Establishment of a tile repository consisting of anchor and receptor-binding domain (RBD) tiles, flanked by position tags (P_start_, P_mid_ and P_end_), sourced from the R2 tailocin and phages infecting *Escherichia coli* or *Klebsiella pneumoniae* and cloned in the pVTE entry vector using VersaTile cloning. (**C**) The use of VersaTile assembly to combine the anchor and RBD tiles in a predefined order in the expression vector. (**D**) The production of engineered tailocins in the producer host PAO1 Δ*prf15* upon induction of the SOS response with mitomycin C to produce the RBP-deficient mutant R2Δ*prf15* and in trans expression of the chimeric RBP with IPTG. (**E**) Results of the spot assay with visible zones at different concentrations of the spotted R2 wild-type (R2-WT) tailocin and its derivates, namely with the in trans expression of the chimeric RBP (R2-WT-trans), the in trans expression of the VersaTile assembled RBP (R2-WT-VT), and the expression of the deficient RBP (R2Δ*prf15*). (**F**) Overview of the results of the spot assay for all R2 wild-type tailocin derivatives. Lower concentrations indicate a higher R2 tailocin activity, and the lowest concentrations at which visible clearance was observed with the naked eye on a bacterial lawn of the *P. aeruginosa* target strain CF510 are displayed.

**Figure 2 antibiotics-14-00104-f002:**
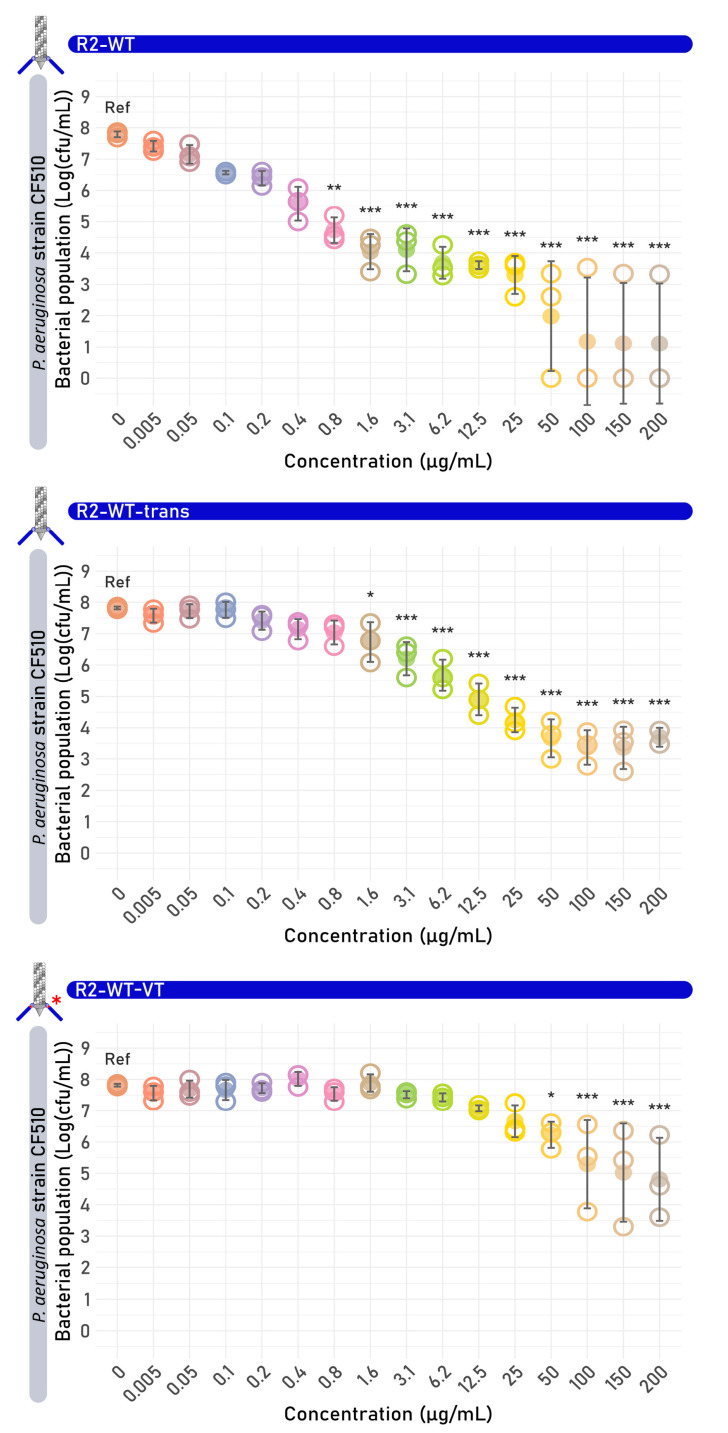
Results of the survival assay of the R2 wild-type tailocin derivatives R2-WT, R2-trans, and R2-VT. The red asterisk of R2-WT-VT indicates the presence of the six-nucleotide junction between the anchor and RBD introduced by the VersaTile technique. Each R2 tailocin was tested on the susceptible target strain *P. aeruginosa* CF510. Significant differences are shown by asterisks (* *p* < 0.05; ** *p* < 0.005, *** *p* < 0.001). Each graph presents the bacterial colony count in function of the concentration of the added R2 tailocin (derivative). The value of each biological replicate is displayed using open circles and the mean values are shown as full circles.

**Figure 3 antibiotics-14-00104-f003:**
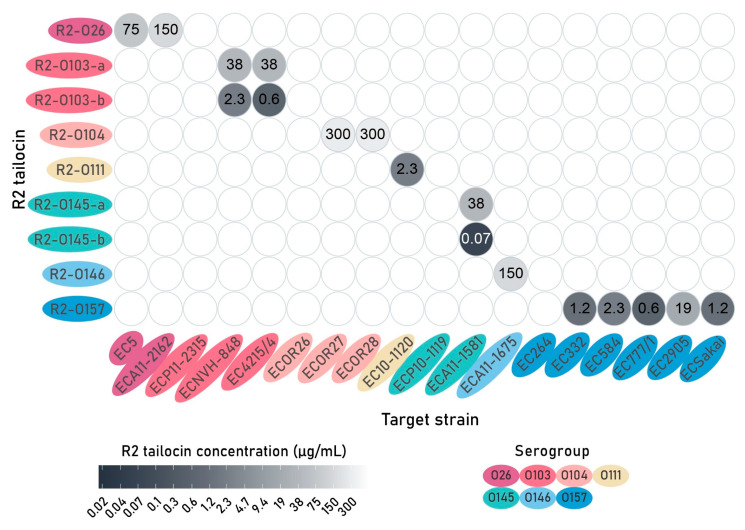
Spot assay of all engineered R2 tailocins tested against all available *Escherichia coli* strains of different serogroups. The lowest concentrations at which clearance was observed on bacterial lawns of *E. coli* target strains are displayed. Lower concentrations indicate a higher R2 tailocin activity. The engineered R2 tailocins and target strains are organized and colored according to O-antigen serogroups of the phage host donating the RBD and the target *E. coli* strain.

**Figure 4 antibiotics-14-00104-f004:**
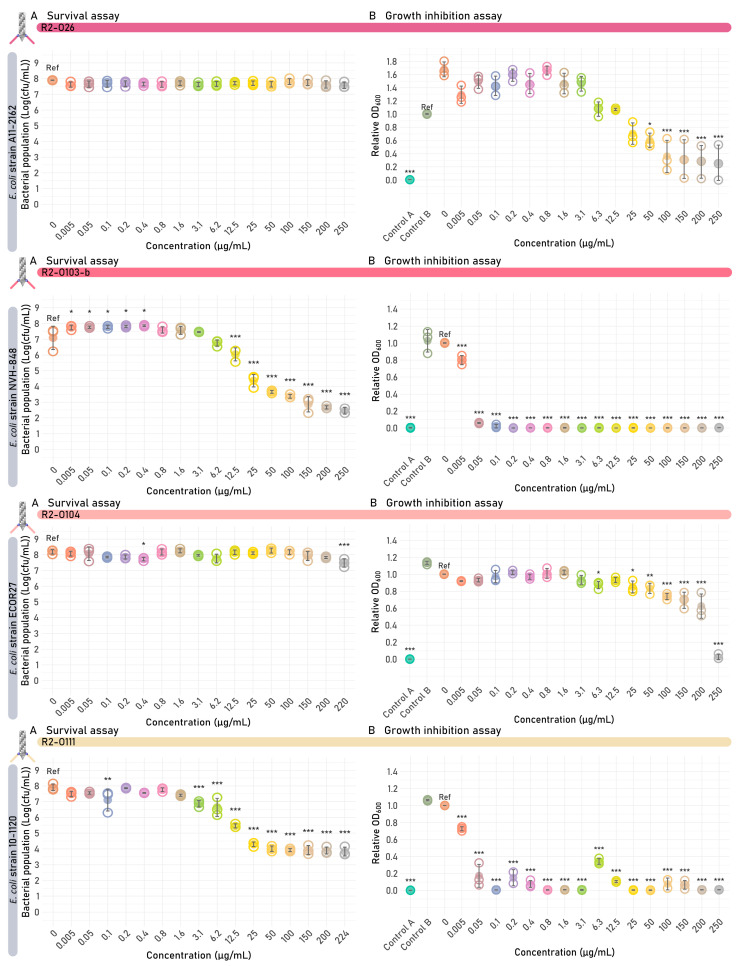
Survival and growth inhibition assays of the different engineered *Escherichia coli* O-antigen-targeting R2 tailocins. One example of each targeted O-antigen serotype is given in this figure. A full version of this figure covering all strains tested can be found in Appendix A. Each R2 tailocin was tested on their susceptible *E. coli* target strains. Significant differences are shown by asterisks (* *p* < 0.05, ** *p* < 0.005, *** *p* < 0.001). Reference (ref) indicates the value that was used as a reference for statistical comparison (untreated sample or control B). (**A**) Results of the survival assay. One plot is shown for each R2 tailocin construct, showing the bacterial colony count in function of the concentration of the added R2 tailocin. The value of each biological replicate is displayed using open circles and the mean values are shown as filled circles. (**B**) Growth inhibition assay results at 8 h are shown per R2 tailocin construct. The relative OD_600_ of each biological replicate is displayed using open circles, and the mean relative OD_600_ is shown as full circles. Two additional controls were performed, one containing the R2 tailocin but without the bacterial strain (Control A) and one containing a receptor-binding protein (RBP)-lacking mutant R2 tailocin particle (R2Δ*prf15*) (Control B). Both controls were added at the highest available R2 tailocin concentration (220–250 µg/mL).

**Figure 5 antibiotics-14-00104-f005:**
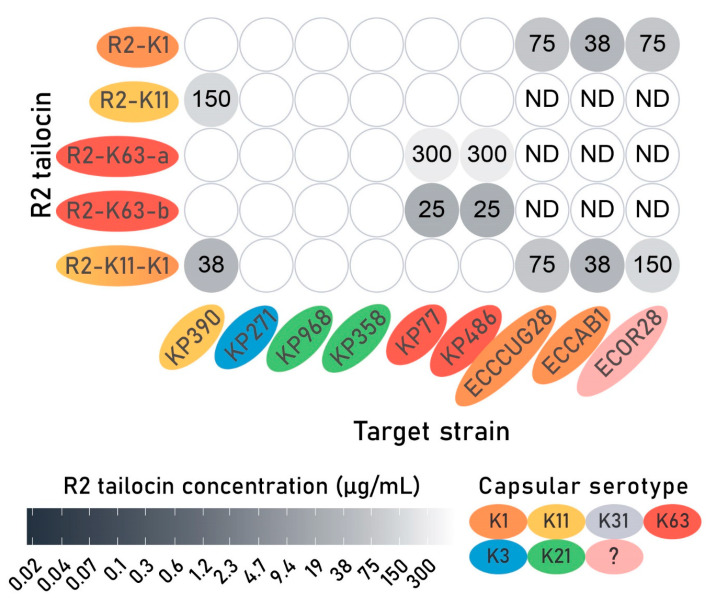
Spot assay of all engineered capsule-targeting R2 tailocins tested against *Klebsiella pneumoniae* and *Escherichia coli* capsular serotypes. The lowest concentrations at which clearance was observed on bacterial lawns of *K. pneumoniae* and *E. coli* target strains are displayed. Lower concentrations indicate a higher R2 tailocin activity. Engineered R2 tailocins that were not spotted against certain strains were indicated as not determined (ND). *E. coli* strain ECOR28 has an unknown capsular serotype, as indicated by a question mark. The engineered R2 tailocins and target strains are organized and colored according to K-antigen serogroups of the phage host donating the RBD and the target *K. pneumoniae* or *E. coli* strain.

**Figure 6 antibiotics-14-00104-f006:**
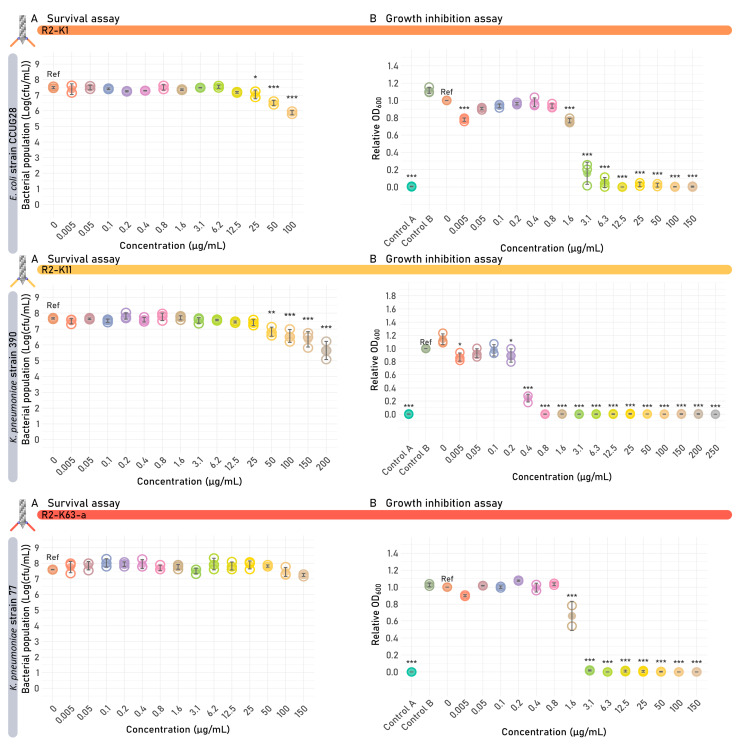
Survival and growth inhibition assays of the capsule-targeting R2 tailocins. One example of each targeted capsular serotype is given in this figure. A full version of this figure covering all strains tested can be found in Appendix A. Each R2 tailocin was tested on its susceptible *E. coli* and *K. pneumoniae* target strain. Significant differences are shown by asterisks (* *p* < 0.05, ** *p* < 0.005, *** *p* < 0.001). Reference (ref) indicates the value that was used as a reference for statistical comparison (untreated sample or control B). (**A**) Results of the survival assay. One plot is shown for each R2 tailocin construct, showing the bacterial colony count in function of the concentration of the added R2 tailocin. The value of each biological replicate is displayed using open circles and the mean values are shown as full circles. (**B**) Results of the growth inhibition assay at 8 h are shown per R2 tailocin construct. The relative OD_600_ of each biological replicate is displayed using open circles, and the mean relative OD_600_ is shown as full circles. Two additional controls were performed, one containing R2 tailocin but lacking the bacterial strain (Control A) and one containing an RBP-lacking mutant R2 tailocin particle (R2Δ*prf15*) instead of the engineered R2 tailocin of interest (Control B). Both controls were added at the same concentration as the highest available R2 tailocin concentration (100–250 µg/mL).

**Figure 7 antibiotics-14-00104-f007:**
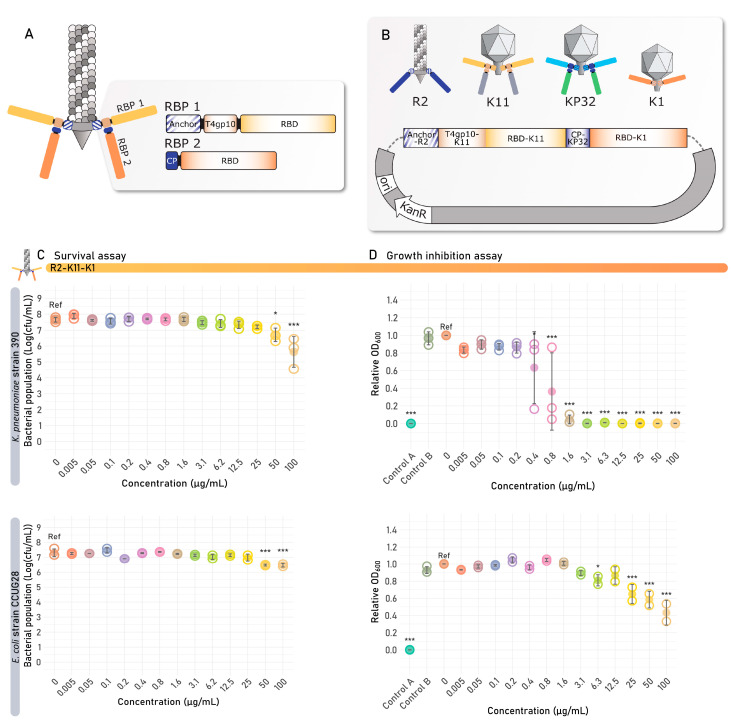
Analysis of bivalent R2 tailocin R2-K11-K1. (**A**) Modular build-up of R2 tailocin R2-K11-K1 with a dual receptor-binding protein (RBP) system with a branched anchor structure. The branched RBP system consists of four building blocks: (I) the N-terminal R2 anchor for attachment of the tailocin baseplate; (II) the RBD with specificity towards *Klebsiella* capsular serotype K11, sourced from phage K11gp17, including the T4gp10-like branching domain; (III) the conserved peptide (CP) sourced from phage KP32 to attach the second RBP to the branching domain of the first RBP; and (IV) the K1 RBD sourced from phage K1F. (**B**) The four tiles were assembled in the shuttle expression vector pVTD29 in a predefined order using VersaTile. (**C**) Results of the survival assay. The bacterial colony count is shown in function of the concentration of the added R2 tailocin. The value of each biological replicate is displayed using open circles and the mean values are shown as filled circles. (**D**) Growth inhibition assay results at 8 h, of the engineered bivalent R2 tailocin targeting both *K. pneumoniae* and *E. coli* capsular serotypes. The relative OD_600_ of each biological replicate is displayed using open circles and the mean relative OD_600_ is shown as full circles. Two additional controls were performed, one containing R2 tailocin, but lacking the bacterial strain (Control A), and one containing an RBP-lacking mutant R2 tailocin particle (R2Δ*prf15*) (Control B). For both the survival and growth inhibition assays, the R2 tailocin was tested on its susceptible *E. coli* and *K. pneumoniae* target strains, which are indicated in vertical, gray-colored headings on the left side of the figure. Significant differences are shown by asterisks (* *p* < 0.05, *** *p* < 0.001). Reference (ref) indicates the value that was used as a reference for statistical comparison (untreated sample or control B).

**Table 1 antibiotics-14-00104-t001:** Overview of all survival and growth inhibition assays of the wild-type and engineered R2 tailocins. Four values are listed per R2 tailocin (derivative) obtained for their corresponding strain(s) tested: (I) log reduction in the cell count (log (CFU/mL)) at an R2 tailocin concentration of 100 µg/mL in the survival assay; (II) the lowest R2 tailocin concentration giving a significant (*p* < 0.05) difference in bacterial count compared to the untreated sample in the survival assay; (III) the lowest R2 tailocin concentration giving a significant (*p* < 0.05) impact on the growth curve at the time point 8 h compared to the untreated sample in the growth inhibition assay; (IV) the minimum inhibitory concentration (MIC) at which >90% inhibition of the growth curve was observed after 24 h. ‘No’ indicates that there was no significant impact on the growth curve or no MIC observed. Not applicable (NA) indicates that the assay was not applicable for the R2 tailocins in question due to the influence of controls. The engineered R2 tailocins are organized and colored according to the target receptor(s) of their RBD(s). ^1^ The negative control containing R2Δ*prf15* (Control B) was used as a reference for statistical analysis instead of the untreated sample. ^2^ This concentration was the lowest tested concentration.

R2 Tailocin Name	Bacterial Strain	Log Reduction in CFU/mL at 100 µg/mL	Minimum R2 Tailocin Concentration (µg/mL)		
			Survival	Impact on Growth Curve at 8 h	MIC
R2-WT	*P. aeruginosa* CF510	6.6 ± 2.0	0.8	-	-
R2-WT-trans	*P. aeruginosa* CF510	4.5 ± 0.6	1.6	-	-
R2-WT-VT	*P. aeruginosa* CF510	2.5 ± 1.4	50	-	-
R2-O26	*E. coli* 5	No	No	6.3	250
	*E. coli* A11-2162	No	No	50 ^1^	250
R2-O103-a	*E. coli* 4215/4	No	250	6.3	200
	*E. coli* NVH-848	No	250	6.3	150
R2-O103-b	*E. coli* 4215/4	3.4 ± 1.5	12.5	150	250
	*E. coli* NVH-848	3.7 ± 0.8	12.5	0.005 ^2^	1.6
R2-O104	*E. coli* ECOR27	No	220	25	No
R2-O111	*E. coli* 10-1120	4.0 ± 0.2	3.1	0.005 ^2^	0.1
R2-O145-a	*E. coli* A11-1581	No	200	6.3 ^1^	250
R2-O145-b	*E. coli* A11-1581	4.6 ± 0.3	1.6	0.005 ^1,2^	0.4
R2-O146	*E. coli* A11-1675	No	150	150 ^1^	200
R2-O157	*E. coli* 264	No	150	100	0.1
	*E. coli* 332	3.0 ± 0.4	100 ^2^	100^2^	100 ^2^
	*E. coli* 584	3.8 ± 0.7	100 ^2^	100^2^	100 ^2^
	*E. coli* 777/1	3.0 ± 0.4	3.1	6.3	No
	*E. coli* Sakai (stx-)	3.9 ± 0.4	12.5	0.005 ^2^	250
R2-K1	*E. coli* CCUG28	1.6 ± 0.2	25	1.6	12.5
R2-K11	*K. pneumoniae* 390	1.1 ± 0.4	50	0.2 ^1^	1.6
R2-K63-a	*K. pneumoniae* 77	No	No	1.6	No
R2-K63-b	*K. pneumoniae* 486	No	No	100	200
R2-K11-K1	*K. pneumoniae* 390	2.0 ± 1.0	50	0.4	12.5
	*E. coli* CCUG28	1.4 ± 0.3	50	25	50

## Data Availability

The original contributions presented in this study are included in the article/Appendix A. Further inquiries can be directed to the corresponding author(s).

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
