# Peer review of "A VersaTile Approach to Reprogram the Specificity of the R2-Type Tailocin Towards Different Serotypes of Escherichia coli and Klebsiella pneumoniae"

_antibiotics, 2025, doi:10.3390/antibiotics14010104_

Round 1
Reviewer 1 Report
Comments and Suggestions for Authors
The manuscript by Dams et al introduces advancements in tailocin engineering including a novel dual-RBP system which is able to target different bacterial species at the same time. The study further highlights the anti-bacterial effects of these protein complexes and their application for inactivation of bacteria. The manuscript is, with some exceptions, well written and easy to follow (unclear sections highlighted below). The methodology applied is both scientifically sound and cutting-edge. However, in my eyes the study has one major flaw: the purification and quantification of produced tailocins and the subsequent detailed comparison of bacteriolytic effects of these based on protein concentrations. While the quantitative aspects of the tailocin effects need a major revision, the qualitative aspect of the tailocin effect, on the other hand, is really interesting and delivers new insights into reprogramming of tailocin host range and the possible therapeutic application.
General comments and questions:
· Main problem of the study: According to the methods section the authors purified the recombinant tailocins using rather crude approaches without a removal step for other proteins. Comparisons of activity of the different constructs are based on concentration measurements of AS precipitated expressions. In these precipitates, impurities contribute to total protein concentration. Thus, the subsequent quantification of the tailocins is rather an estimation than a precise measurement as the measured protein concentrations include tailocins and other host cell proteins. Differences in expression level will result in different proportions of tailocins to host cell protein. In summary, activity comparison are only possible if tailocin proportions are identical in all samples. This needs to be proven, otherwise all that comparative data and the conclusions are not valid in my eyes. Variations in inhibitory effect on bacterial growth of the different purification methods emphazise this in my eyes. That also applies to MICs shown in table 1, growth inhibtion curves shown in Figure 2, Figure 3 etc etc. In contrast to selective purification procedures based on affinity ( tags), charge ( Ion exchange) or size ( Size exlusion) alone or in combination, ammonium sulfate precipitation and high speed centrifugation are not selective and are rather be considered as enrichment or extraction procedures. In combination with the procedures listed above they can be named as one purifucation step. As a stand-alone method used in this work the naming "purification" is misleading and not correct in my eyes.
· Is it possible to include a purification control without tailocin or without mitoC induction? I know that this is not too easy, but I think that a control would really help here
· I believe that introducing a purification tag somewhere at the distal part of the baseplate could enable specific affinity based purification of the tailocin and therefore allow for accurate tailocin quantification and thus better comparability between different tailocins and production batches.
· Is it possible to encode all tailocin genes on one big expressions plasmid? This would align RBP production with the rest of the tailocin and also allow to produce the tailocins in a host deficient of other antibacterial compounds such as F-type tailocins.
· What was the purpose of using plasmids with and withough lac repressor? Was the rather leaky expression in lac- plasmid beneficial for tailocin formation compared to the lac+ plasmids?
· Is it possible to include calculations how many protein complexes are necessary to kill one bacterium?
· The figures 2,4,6 depicting the survival- or growth inhibition assays are very large and repetitive. Please choose representative graphs and put the rest into supplement. Or choose a different way of depicting your results.
Specific comments and questions
· Line 48: R2 type tailocins not properly introduced, only R type in general
· Line 64-67: please rephrase, hard to follow. The following lines as well, probably difficult to understand for someone who is not familiar with typical requirements of phage therapy or alternatives antimicrobial approaches.
· Line 66/82: please include 1 or 2 short sentences to elaborate the sur mesure and pre-a-porter paradigms
· Line 90: relies – use past tense
· Line 104/105: please rephrase, I suggest “… have not yet been targeted be tailocins before”
· Section 102-111: please rephrase, the section is hard to follow and confusing as the authors jump between explanatory statements and plans/outlines of this study. This might be even worse for someone who is not “from the field”
· Line 252: chimera
· 393-400 / 505-508: why was the CP part of phage KP32 used to anchor the second RBP (of phage K1) to the first RBP (of phage K11) and not the CP part of phage K11?
· 485-488: Would be beneficial to show structure prediction of the proteins described in this study. This would also give further insights into the chimeric junctions which have been introduced to assemble the dual RBP tailocin
· 560-570: Have the authors thought about applying mass spectrometry to investigate the expression and assembly kinetics of chimeric tailocins? This could most likely answer the questions raised in this section of the discussion
· 599-601: Do the authors have any ideas about why this is the case? (Other than a possible masking of the o-antigen)
Formatting:
There are many errors with formatting throughout the manuscript even though some of the below mentioned issues might have been caused by the journal’s processing of the manuscript:
· Please use different enumeration characters than “(1), (2),…” etc. as the journal uses this style to display the references in the text. E.g.:
o L. 18-21
o L. 97/98
o L. 139/140, 144
· Also, please use them consistently throughout the text as you e.g. changed from “(1),(2)…” to “(I),(II)..” (L. 182/183).
· In L. 60 - L. 71 a different font is used. Please correct
· L. 120: It looks like there is a double space after “Pmid”. Please check.
· L. 216/ L. 235: As the journal always formats the titles in italic “E. coli” has to be written in normal style. Please make a note to the editors.
· L. 289: Within the title: Please change “Pneumoniae” to “pneumoniae”
· Please re-check the links/citations of your figures. Despite the supplementary figures all brackets that should contain the citation of the respective figure state ““Error! Reference source not found”
o e.g.
o L. 123
o L. 155
o L. 171
o L. 187
o L. 188
o L. 238
· Etc.
· L. 57/58: Please only write „RBP“ as the abbreviation has already been introduced in line 45
· L.114/ L. 118: Here, you introduce the abbreviation RBD again (probably because it’s the first time you use it in the results part?) However, you do not introduce the abbreviation RBP again. Please either explain or directly use the short versions to stay consistent.
· Unfortunately, in the PDF I received for revision all figures were really blurry and therefore hard to read. Please re-check the resolution/quality of your uploaded figures.
Reviewer 2 Report
Comments and Suggestions for Authors
Summary:
In the paper “A VersaTile Approach to Reprogram the Specificity of the R2-Type Tailocin Towards Different Serotypes of Escherichia coli and Klebsiella pneumonia” the authors have made a platform for the quick shuffling of receptor binding proteins for engineering the receptor recogniton of tailocins. In initial testing of the VersaTile technique, the authors showed that the engineered tailocins R2-WT-VT did have killing efficiency; however, it was lower than the R2-pyocin WT and the R2-WT-Trans. The authors expanded on the tailocin platform by shuffling enzymatic RBPs from phages that target the polysaccharides of E. coli and Klebsiella. While some of the tested strains needed a high concentration of the tailocins, others were more sensitive. Lastly, the authors created a tailocin with two RBPs which have never been done before. The bivalent tailocin showed higher growth inhibition of the Klebsiella (K11) strain than the E. coli (K1) strain. Overall, the study shows that a VersaTile platform for quick engineering of the tailocin is possible, however, with much less efficiency than the traditional cloning of the RBP (R2-WT-Trans). The study also shows that tailocins can be engineered to target Klebsiella, which has not been done before. Lastly, the most novel part of the study is the creation of a bivalent tailocin that targets both E. coli and Klebsiella.
General concept comments:
In the initial experiment (Figure 2), where the authors test the killing efficiency of the R2-pyocin and their VersaTile platform, a decrease in growth is shown in the negative control B (R2-pyocin R2Dprf15). The pyocin does not express a receptor binding domain; hence it should not bind and kill the cells. The authors suggest that the co-purification of an F-type pyocin from Pseudomonas aeruginosa may impact their results. This raises concerns about the validity of their findings, as a non functional negative control could compromise the reliability of the assay. Additionally, the true killing efficiency of the VersaTile tailocin (R2-WT-VT) might be artificially inflated if the F-type pyocin contributes to these effects. To resolve these issues, the authors should create an F-type pyocin knockout mutant to accurately assess the efficiency of the engineered tailocins.
The negative control (control B) showed killing/growth inhibition in several of the experiments (Figure 4, 6 and Figure S1 and S3): R2-O26 on E. coli strains A11-2162, 215-4 and NVH-848 , R2-O157 on E. coli 777/1 and Sakai, R2-K11 on Klebsiella strain 390, etc. compared to the reference (0 µg/mL). This suggests that control B does have an effect on the growth of the tested strains. In theory, the F-type (that is copurified) and R2-pyocin R2Dprf15 should not be able to kill E. coli and Klebsiella; however, the results suggest otherwise. Again, a knockout of the F-type operon would show if the F-type and/ R2-pyocin R2Dprf15 has any influence on the killing efficiency.
If F-type pyocin particles are present in the final purified engineered tailocins, then the concentration of the engineered tailocins cannot be determined. This further complicates any assessment of the relationship between dose and killing efficiency. Furthermore, can the authors rule out that the inhibition of growth is due to direct killing or due to lysis from outside, especially at the high concentration?
Overall, the validity of the experiments reported in the study is significantly undermined by concerns regarding the negative control (control B) and the possibility of contamination of the F-type pyocin. The presence of killing or growth inhibition in the negative control across multiple experiments compromises the reliability of the findings, as the control should be unable to kill the tested strains. Moreover, the presence of F-type pyocin in the purified engineered tailocins makes it difficult to determine their concentration accurately, complicating the interpretation of the dose-response relationship. The mechanism of growth inhibition—whether due to direct killing by the tailocins or non-specific lysis—remains unresolved, particularly at high concentrations. All of this need to be addressed before this paper can be accepted.
Specific comments:
L31: reference should be provided.
L33-42: It would be helpful to describe the organisms in which tailocins are found.
L52: I think reference 6 is not correct. The sentence refers to the R2 tailocin mode of action, but the reference is about the structure of the R1 and R2 tailocin. The authors should reference the original paper about the mode of action: “Dansyl chloride labeling of Pseudomonas aeruginosa treated with pyocin R1: change in permeability of the cell envelope” or “Pyocin R1 inhibits active transport in Pseudomonas aeruginosa and depolarizes membrane potential”.
L119: Klebsiella and E. coli are not in italic. This issue appears in multiple places and needs to be corrected accordingly.
L80: The references 20 and 21 seem to be wrong. 21: the title of the paper is wrong. 22: the paper is about Innolysins and not pyocins.
Figure 1: should be bigger. It is difficult to see the spots in the figure.
Table 1: Could the author explain the superscript i, in the column about the impact of the growth curve after 8 hrs?
Table 1 and Figure S1: For tailocin R2-O26, the author claims that they observe an impact of growth at the concentration of 6.3 µg/mL however there is no impact on the growth 12.5, 25, 50 and 100 µg/mL? Similar trends can be observed for other tailocins. However, it is very difficult to see the true OD value in the data, so I recommend that the figures should be made bigger, so it is visible to see the OD value after 8 hrs. The negative control seems to have a major impact on the growth in several of the growth inhbition assays. This is not adressed in the paper.
L162-L165: The results shown in Figure 2 are difficult to believe as the negative control also shown a killing effect in the growth inhibition assay. Overall, it is a problem that the F-type pyocin could be present in all the samples.
L218: An introduction to the paragraph would make it easier for the reader to follow the aim of the experiments.
L237: The spot assay results should be provided so the reader can see the clearing zones. Enzymatic RBPs often make a halo morphology. Did the authors also see this in their spot assay?
L242-243: Can the authors rule out that the killing of the bacteria is not due to lysis from without instead of actual killing from the tailocin?
L575: the author says that enzymatic RBPs have antibacterial properties. Does that mean that the RPBs they have chosen to have antibacterial properties? In that case, can it be ruled out that the growth inhibition observed is because of the RBPs on the tailocin and not the true action of the tailocin?
L596-604: A host range analysis of the phages could be beneficial to understanding the work produced in this study. It could be because there are internal defense systems, but an adsorption assay would show if the phage binds to the polysaccharides. It could be that the RBP does not recognize the polysaccharides. It has been shown before that O-antigens can be modified, masking it as a receptor for RBPs. Thus, purified RPBs spotted on the lawn could show if the RBP binds and degrade the polysaccharide. This would explain why some of the tailcoin does not kill some of the tested strains. In the method section it is stated that some of the phages were spotted in on the strains but the results is not shown?
Reviewer 3 Report
Comments and Suggestions for Authors
Overall this manuscript is well written and capably presented. I have only a few minor editing changes I would suggest.
1) throughout the manuscript the message: (Error! Reference source not found. Please fix the problem for these. Also there is a change in font from lines 60-70.
2) The authors use the format of (1) for citations but also they use the same format for ordering points (lines 18, 97, 139, 684) and similarly for 2, 3, etc.
3) It would be nice to have a table that lists all the final protein sequences of their constructs. This could be in supplemental.
4) Given that many of the constructs they created had little or no killing activity, are there any general principles the authors have learned from which ones work and which ones don't work? This would be nice in the discussion alongside the section on Technical Aspects of Engineered Tailocins Production Impacting the Final Antibacterial Effectiveness
5) There are some incomplete citations (see 10, 15, 20, 22, 23, 24, ...) I've not listed them all so please review each carefully.
6) At the very end are 3 footnotes, but these will never be seen by readers in that location. Consider incorporating the as parenthetical clauses in the text or figure legends.
Round 2
Reviewer 1 Report
Comments and Suggestions for Authors The revised manuscript presents a substantial improvement in clarity and accessibility, particularly for readers who may not be specialists in the field. The authors have provided thorough and thoughtful responses to my questions, which not only address my concerns but also offer valuable background information. This additional context has greatly enhanced my understanding of their experimental design and overarching research strategy. Furthermore, I appreciate how the authors have incorporated the suggestions made by Reviewer 2 and myself, which have collectively contributed to refining the manuscript. The revisions have not only clarified the methodology but also strengthened the overall narrative of the research. In light of these improvements and the authors' diligent efforts to address the feedback provided, I strongly recommend that the manuscript be accepted for publication. It represents a significant contribution to the field and will undoubtedly benefit a wide audience.Reviewer 2 Report
Comments and Suggestions for Authors
I want to say thank you to the authors for answering my concerns. I believe they have adressed everything and make the reccomended adjustments. The additions to the new manuscript are sound and elevate it. The new statistical method is more reliable, and removing the growth inhibition assay of the wild-type tailocins is a good decision. I do not have any comments except that the authors should remove the (Figure 2, a) and Figure 2, b) in L180 and L267, respectively, as there are no a and b in figure 2.